# Preparation and Properties of Salecan–Soy Protein Isolate Composite Hydrogel Induced by Thermal Treatment and Transglutaminase

**DOI:** 10.3390/ijms23169383

**Published:** 2022-08-20

**Authors:** Jing Gan, Lirong Sun, Chenxia Guan, Teng Ren, Qinling Zhang, Shihui Pan, Qian Zhang, Hao Chen

**Affiliations:** 1College of Life Science, Yantai University, No. 30 Qingquan Road, Laishan Strict, Yantai 264000, China; 2Marine College, Shandong University, No. 180 Wenhua West Road, Gao Strict, Weihai 264209, China

**Keywords:** salecan, soy protein isolate, composite hydrogel, mechanical properties, rheological properties

## Abstract

Salecan (Sal) is a novel marine microbial polysaccharide. In the present research, Sal and soy protein isolate (SPI) were adopted to fabricate Sal–SPI composite hydrogel based on a stepwise process (thermal treatment and transglutaminase induction). The effect of Sal concentration on morphology, texture properties, and the microstructure of the hydrogel was evaluated. As Sal concentration varied from 0.4 to 0.6 wt%, hydrogel elasticity increased from 0.49 to 0.85 mm. Furthermore, the internal network structure of Sal–SPI composite hydrogel also became denser and more uniform as Sal concentration increased. Rheological studies showed that Sal–SPI elastic hydrogel formed under the gelation process. Additionally, FTIR and XRD results demonstrated that hydrogen bonds formed between Sal and SPI molecules, inferring the formation of the interpenetrating network structure. This research supplied a green and simple method to fabricate Sal–SPI double network hydrogels.

## 1. Introduction

Hydrogel is an extremely hydrophilic polymer material with a three-dimensional network structure [1]. As an excellent water absorption and water retention material, hydrogel has attracted considerable attention in the past few years and has been widely used in the fields of food [2,3], cosmetics [4], tissue engineering [5], and drug delivery [6] as a typical natural polymer.

Proteins and polysaccharides have been widely utilized in food hydrogel systems. Due to their low toxicity, biocompatibility, and biodegradability, the studies on polysaccharide (sodium alginate, gellan gum, carrageenan, chitosan, dextran, etc.) and protein (soy protein isolate (SPI), pea protein, whey protein, zein, etc.) hydrogel systems have been increasing in recent years [7,8]. However, single protein hydrogels usually have poor gel strength, water holding capacity, and thermal stability, while the polysaccharides usually show high viscosity and poor solubility, which limits the application of protein and polysaccharide. Therefore, the focus of protein and polysaccharide hydrogel research is changing from single hydrogels to composite hydrogels [9,10]. Polysaccharides and proteins can be compounded and cross-linked by chemical reagent cross-linking [11,12], enzyme induction [13], Schiff base reaction [11], etc., to construct composite hydrogel system to improve the hydrogel properties. A briefly summary of gelling methods and characteristics of protein–polysaccharide composite hydrogel was shown in Appendix A [11,12,13,14,15].

Polysaccharide–protein composite hydrogels have excellent mechanical properties, rheological properties, and water retention properties compared to single hydrogels [16,17]. The protein network can be considered a flexible backbone to improve the elasticity of the hydrogel, while the polysaccharide network would exhibit a rigid backbone to increase the stiffness of the hydrogel. In addition, the microstructure of composite hydrogels can be altered by changing internal or external gelation factors, such as the ratio of polysaccharides to proteins, cross-linking agents, and the environmental pH, to obtain the desired hydrogel structure, which also facilitates the regulation of functional factor release. Zhou et al. [18] prepared konjac glucomannan (KGM)–fish gelatin (FG) composite hydrogels through alkali and heat treatment, which significantly improved the properties of single KGM hydrogels with fast water evaporation rate and slow degradation. Some other investigators found that when SPI was combined with corn fiber gum, the composite hydrogel exhibited denser and more regular inner structure [19].

Salecan (Sal) is a novel water-soluble β-glucan harvested from the fermentation medium of Agrobacterium sp. ZX09 and it is composed of D-glucose units connected by α-(1–3) and β-(1–3) glycosidic linkages [20]. The presence of many derived functional groups, such as hydroxyl groups, on Sal backbone allows for easy chemical modification [20]. Furthermore, Sal solutions show high viscosities with low concentrations, exhibiting typical viscoelastic and non-Newtonian fluid properties. The temperature (<55 °C) and pH (6.0–12.0) have a slight effect on the viscosity of Sal solutions [21]. The strong hydrophilicity, good rheological properties, and biodegradability of Sal make it an ideal material for the manufacture of hydrogels [20,22]. Sal has been successfully used for cell adhesion, wastewater treatment, drug delivery, etc. [23,24,25]. In addition, Sal has been reported in many biological activities, such as lowering blood lipids, lowering blood glucose, promoting weight loss, relieving constipation symptoms, increasing intestinal fatty acid and probiotic levels, reducing acute liver injury, etc. [26,27]. According to several studies, Sal-based hydrogels usually required toxic chemical agents such as N, N′-methylene diacrylamide and N′-isopropylacrylamide to engage in composite hydrogel networks [28,29], which have limited their applicability. Therefore, it is necessary to explore green methods to construct Sal-based hydrogels. In our previous research, ions, freeze-thawing, and thermal treatment were adopted to induce Sal to form hydrogel, and thermal treatment worked [30]. However, the springiness of Sal hydrogel needed to be improved.

SPI performs excellent emulsification and stabilization, while the weak intermolecular interactions between polypeptide chains contribute to the poor mechanical and water barrier properties of SPI-based hydrogel [31]. Complexation of polysaccharides with SPI could improve the functionality of SPI hydrogels. Liu et al. [32] prepared a robust and resilient SPI/oxidized dextran hydrogel. Oxidized dextran, as a cross-linking agent, significantly improved the mechanical properties and stability of SPI. It was also found that the mechanical strength of composite hydrogels formed by agar and polyethylene glycol (PEG)-modified SPI was greater than that of single SPI hydrogels [33]. Therefore, the combination of Sal and SPI has potential to form a stable composite hydrogel while retaining the advantages of SPI and Sal.

In this work, the Sal–SPI composite hydrogel system was first fabricated by thermal treatment and TGase induction. The effect of Sal concentration on the apparent morphology, texture characteristics, swelling behavior, and internal morphology of Sal–SPI composite hydrogel was evaluated. Subsequently, the gelation process of Sal–SPI composite hydrogel was monitored by rheological studies, and cross-linking mechanisms were investigated through Fourier transform infrared spectroscopy (FTIR) and X-ray diffraction (XRD). This research might provide a novel composite hydrogel and broaden the application range of Sal and SPI in the industry.

## 2. Results and Discussions

### 2.1. The Effect of Sal Concentration on the Apparent State of Sal–SPI Composite Hydrogel

Figure 1a shows the state of different composite hydrogel precursor solutions (SS0, SS4, SS5, SS6, SS7, SS8) after thermal treatment. The hydrogel precursor solutions were homogeneous and stable when Sal concentration ranged from 0.4 to 0.6 wt%. However, when Sal concentration exceeded 0.7 wt%, the solution began to show obvious stratification.

According to Figure 1b, SPI single hydrogel (SS0) was soft and exhibited poor moldability. As Sal concentrations increased from 0.4 to 0.6 wt%, the moldability of composite hydrogels was enhanced. When Sal concentration increased above 0.7 wt%, the firmness of the composite hydrogel began to deteriorate.

The above phenomenon can be explained by the homogeneity of the composite hydrogel precursor solution. When precursor solutions were consistent, Sal and SPI could be cross-linked with each other to generate Sal–SPI composite hydrogels with good moldability after stepwise induction. When the ratio of Sal and SPI exceeded a certain critical value (Sal concentration is 0.7 wt%), the polysaccharide and protein in the hydrogel system could undergo phase separation due to thermodynamic incompatibility. There was a competition between the gelation process and the phase separation process. The moldability of Sal–SPI composite hydrogel deteriorated when the Sal–SPI system was based on the phase separation process.

### 2.2. The Effect of Sal Concentration on Water Holding Capacity (WHC) and Water Content (WC) of Sal–SPI Composite Hydrogel

The WHC and WC of Sal–SPI composite hydrogel obtained by compounding 5.0 wt% SPI with different concentrations of Sal are shown in Figure 2. During food preparation and storage, single protein hydrogels are prone to flocculation and aggregation, lowering the WHC of hydrogels [8]. Because of the interaction forces such as hydrogen bonds, van der Waals forces, and electrostatic interactions between polysaccharides and proteins, the addition of polysaccharides could improve the WHC of single protein hydrogels. As shown in Figure 2a, all Sal–SPI composite hydrogels had high WHC values above 80%. In Figure 2b, all Sal–SPI composite hydrogels showed high WC values exceeding 90%. Sal is a highly hydrophilic chemical that is rich in hydroxyl groups and it can improve the composite hydrogel’s ability to bind additional water molecules. However, when Sal concentration was greater than 0.7 wt%, the WHC and WC of composite hydrogel both decreased, suggesting that slight phase separation between Sal and SPI occurred, destroying the internal network structure of the hydrogel and reducing WHC and WC [34]. 

### 2.3. The Effect of Sal Concentration on the Swelling Behavior of Sal–SPI Composite Hydrogel

The swelling behavior of hydrogel affects its role as a delivery vehicle and its degree of digestion in the gastrointestinal tract [22]. Figure 2c shows the Equilibrium swelling degree (ESD) of Sal–SPI composite hydrogel with varied Sal concentrations. With the increase of Sal concentration, the ESD of Sal–SPI composite hydrogel increased initially, then decreased. The surface hydrophilicity and overall density of the network matrix are both important factors in the hydrogel’s capacity to swell. When Sal concentration increased from 0.4 to 0.5 wt%, the ESD of the composite hydrogel increased from 702% to 767%. This might be due to the strong hydrophilicity of the multiple hydroxyl groups on the Sal backbone. The water affinity of Sal–SPI composite hydrogel rose as Sal concentration increased, promoting water molecule penetration into the hydrogel framework and contributing to the hydrogel having higher hydration [24]. In addition, there were more cross-linking points between Sal and SPI as Sal concentration increased, contributing to the ESD rising. In addition, the segment relaxation factor also has an impact on the swelling process of the hydrogel. When Sal concentration continued to rise to 0.6 wt%, the ESD of Sal–SPI composite hydrogel began to diminish. This could be that as Sal concentration increased, the cross-linking points on the molecular chain of Sal and SPI increased, shortening the distance between adjacent cross-linking points, slowing the rate of relaxation of the chain segments between the cross-linking points, and lowering the ESD of the composite hydrogel [18]. When the concentration of Sal increased to 0.7 wt%, the ESD of the composite hydrogel reduced dramatically to 614%. Protein and polysaccharide are macromolecular polymers, and maintaining a balance of neither repulsion nor attraction between the two becomes increasingly difficult as their concentration increases, resulting in phase separation [35]. 

### 2.4. The Effect of Sal Concentration on the Texture Properties of Sal–SPI Composite Hydrogel

The texture parameters of the composite hydrogel produced by different concentrations of Sal and 5.0 wt% SPI are shown in Table 1. With the increase of Sal concentration, all the texture characteristics of Sal–SPI composite hydrogel showed a tendency of first rising (SS4–SS6) and then declining (SS7).

The texture properties of Sal–SPI composite hydrogel might improve as Sal concentration increased because: (1) Sal, as a polysaccharide with high molecular weight, exerted a space-occupying effect on SPI hydrogel. The addition of Sal to the hydrogel system could significantly improve the concentration of SPI, enabling the creation of the hydrogel with greater viscoelasticity [36]. (2) Sal, being an anionic polysaccharide, could interact with positively charged ions such as amino groups in SPI by negatively charged carboxyl groups under certain conditions. As Sal concentration increased, the electrostatic interaction between Sal and SPI was enhanced to promote cross-linking with each other, thereby enhancing the texture characteristics of Sal–SPI composite hydrogel. (3) Sal is rich in hydroxyl groups and could form a physically cross-linked hydrogel with the amino group of SPI through hydrogen bonding. When Sal concentration increased to 0.7 wt%, the textural indexes of the composite hydrogel began to decrease. This may be due to the minor phase separation between Sal and SPI, which caused the mechanical properties of the composite hydrogel performance to deteriorate.

### 2.5. The Effect of Sal Concentration on the Internal Structure of Sal–SPI Composite Hydrogels

The influence of Sal concentration on the internal structure of Sal–SPI composite hydrogel can be observed by Scanning electron microscopy (SEM) with a magnification of 600× (Figure 3a). As shown in Figure 3a, there was no cross-linked network detected inside, indicating that SPI cannot form an elastic hydrogel through thermal induction at this concentration. In Figure 3b, a cross-linked network appeared inside the hydrogel, but the network was messy and the pore sizes were uneven. As Sal concentration increased, the hydrogel network became denser and more regular, and the pore size became more homogeneous. The internal structure of Sal–SPI composite hydrogel with a Sal concentration of 0.6 wt% was the most homogeneous and densest, as illustrated in Figure 3d. This phenomenon further supported the macroscopic indicators of the composite hydrogel. This phenomenon could be explained by the fact that as the amount of Sal that was added increased, there were more cross-linking points between Sal and SPI, resulting in a higher cross-linking density.

However, when Sal concentration reached 0.7 wt% (Figure 3e), the network became messy and irregular, and the pore diameters increased. The internal structure was no longer a typical hydrogel network structure until the concentration rose to 0.8 wt% (Figure 3f), indicating that it was no longer a typical elastic hydrogel. In this system, the phase separation and gelation process were in a competitive interaction. The gelation process of the two were affected by phase separation, which weakens the hydrogel structure [37]. In addition, due to the hydrophilicity of Sal, the hydrogel system absorbed more water. During the freeze-drying process, the ice crystals were formed in the hydrogel then sublimated to generate huge cavities in the hydrogel finally [38].

### 2.6. The Rheological Properties of Sal–SPI Composite Hydrogel

Figure 4a,b shows the single gelation processes of 0.6 wt% Sal and 5.0 wt% SPI, respectively, to determine whether the two can form a single hydrogel network at this concentration. It can be figured out that during the whole process of heating and cooling of Sal, the storage modulus G′ was always smaller than the loss modulus G″. Although G′ and G″ briefly crossed during heating at 90 °C, in the subsequent cooling process, G′ was smaller than G″ throughout the subsequent cooling process, demonstrating that 0.6 wt% Sal cannot form a single network hydrogel through heating. The gelling process of 5.0 wt% SPI is shown in Figure 4b. It was found that SPI cannot form a single network hydrogel by TGase induction.

The gelation process of Sal plus TGase and the thermal treatment process of SPI were explored to rule out the influence of TGase on the gelation of Sal and the influence of thermal treatment on the gelation of SPI. As shown in Figure 4c, The G″ of Sal was always greater than G′. This result revealed that adding TGase to 0.6 wt% Sal did not promote the formation of Sal single hydrogel. In addition, in Figure 4d, the results also showed that during the thermal treatment, G″ was always greater than G′. Therefore, SPI also cannot be formed into a single hydrogel by heating and cooling at this concentration.

In conclusion, neither 0.6 wt% Sal nor 5.0 wt% SPI could form a single hydrogel network, and their induction methods had no effect on the hydrogel processes of each other. Then 0.6 wt% Sal and 5.0 wt% SPI were mixed to investigate whether the two can form a composite hydrogel by thermal treatment and enzyme induction. As shown in Figure 4e, the G′ and G″ of the mixed system began to intersect during the heating process at 90 °C, and G′ > G″ in the final cooling process. There was no significant difference between G′ and G″, indicating that the composite hydrogel was a weak hydrogel. This phenomenon could be caused by the melting of Sal and SPI molecular chains at a high temperature of 90 °C, resulting in random chains. At high temperature, the movement of single-stranded molecules accelerated, and the hydroxyl groups of Sal and the amino groups of SPI could generate intramolecular and intermolecular hydrogen bonds, causing the molecular chains to become entangled, but the concentration was too low to form enough hydrogen bonds to maintain a stable network structure, resulting in a weak hydrogel.

Figure 4f shows the gelation process of the heated and cooled mixture after adding the TGase. The initial difference between G′ and G″ was quite tiny, as can be seen from the temperature–time scanning curve, indicating that the fragile hydrogel structure generated in the heating and cooling process was destroyed after introducing the enzyme. The TGase has high enzymatic activity at 50 °C, allowing it to catalyze the cross-linking interaction between the ε-amino group on the lysine residue in SPI molecule and the γ-hydroxyamide group on the glutamine group. Furthermore, the hydroxyl group of Sal can cross-link with the amino group of SPI by forming hydrogen bonds. The hydrogel system will be sturdier and stiffer because of the thickening effect of Sal. As shown in Figure 4f, the G′ of the Sal–SPI system was much greater than G″, which further confirmed the formation of Sal–SPI composite elastic hydrogel.

In summary, the step-by-step induction method was discovered to be able to cross-link Sal and SPI, which could not form a single hydrogel originally.

### 2.7. Spectrum of Sal–SPI Composite Hydrogel

#### 2.7.1. FTIR

The infrared spectra of Sal single hydrogel, SPI single hydrogel, and Sal–SPI composite hydrogel are shown in Figure 5a. It can be seen that Sal has a strong characteristic peak at 3424 cm^−1^ of -OH stretching vibration [39]. The characteristic peak centered at 1038 cm^−1^ is likely to correspond to the -OH stretch of the glucopyranose ring. The weak band at 809 cm^−1^ is due to the presence of α-glucopyranose in the polysaccharide chain, while the band at 891 cm^−1^ indicates that D-glucopyranose is connected by β-configuration [40]. In the fingerprint area of the infrared spectrum of Sal–SPI composite hydrogel, there are characteristic absorption peaks of α-glucopyranosyl and β-glucopyranosyl units (869 cm^−1^, 810 cm^−1^), as well as the amide I and II area characteristic absorption bands of SPI: the peaks located near 1612 cm^−1^ and 1529 cm^−1^ are related to the C-O stretching in the amide I region and the N-H bending vibration in the amide II region, respectively. Compared with the single SPI hydrogel, the -OH peak of Sal–SPI composite hydrogel shifts from a low to a high wavenumber (from 3424 cm^−1^ to 3433 cm^−1^), indicating that the -OH peak is blue-shifted. It can be speculated that the interchain hydrogen bonds between Sal and SPI cause the cross-linking reaction between Sal and SPI, resulting in the formation of a Sal–SPI composite hydrogel. In addition, thermogravimetric results (Appendix A) showed that the thermal stability of Sal–SPI composite hydrogel improved compared to the SPI and Sal which might due to the interaction forces between SPI and Sal [22,41]. The gelling mechanism of Sal–SPI composite hydrogel is shown in Figure 1.

#### 2.7.2. XRD

Figure 5b shows the XRD spectra of Sal, SPI, and Sal–SPI composite hydrogels. Sal features a diffraction peak approximately 2θ = 21.2°, which corresponds to the interaction between the partial functional groups of Sal chain and is a typical absorption peak of the semi-crystalline polysaccharide structure of Sal. Similar diffraction peaks are also detected in other polysaccharides such as carboxymethyl cellulose [42]. SPI has two diffraction peaks near 2θ = 9.4° and 20.1°, which may be due to the crystalline domain formed by the interaction of some functional groups of the protein, such as hydroxyl and amino groups. In the Sal–SPI composite hydrogel, the intensities of the two diffraction peaks that are near 2θ = 9.4° and 20.1° are significantly increased, indicating that Sal and SPI may be cross-linked through the interaction force. The crystallinity of the composite hydrogel system is improved when compared to the single SPI hydrogel system, and a more regular molecular arrangement is generated.

### 2.8. The Biodegradability of Sal–SPI Composite Hydrogel

The biodegradability of hydrogel is an important factor to evaluate the application performance of the Sal–SPI composite hydrogel in the food industry. As shown in Figure 6, the weight loss of the Sal–SPI composite hydrogel increased with time. When the degradation time was 60 h, the remaining weight of the Sal–SPI composite hydrogel was only 8.79% of the initial value. This phenomenon shows that the Sal–SPI composite hydrogel has excellent biodegradable properties.

## 3. Materials and Methods

### 3.1. Materials

Salecan (Mw~2000 kDa, 95.0% purity) was obtained from Synlight Biotech Co., Ltd. (Chengdu, China). Soy protein isolate was purchased from Yuwang Industrial Co., Ltd. (Dezhou, China). TGase was supplied by YiMing Biological Techology Co., Ltd. (Taizhou, China). Other chemicals were analytical grade.

### 3.2. Preparation of Sal–SPI Composite Hydrogel

SPI solution (5.0 wt%) was prepared in deionized water by stirring for 3 h at room temperature and stored at 4 °C overnight. Then certain amounts of Sal power (0, 0.4 wt%, 0.5 wt%, 0.6 wt%, 0.7 wt%, 0.8 wt%; SS0, SS4, SS5, SS6, SS7, SS8) were mixed with SPI solution under stirring for 6 h at room temperature and stored at 4 °C overnight. The solution of Sal–SPI mixture was preheated on a water bath (90 °C) for 20 min and immediately cooled down to room temperature by running water. Then TGase (20 U/g) was added to the solution of Sal–SPI mixture and cultured at water bath at 50 °C for 1 h to obtain Sal–SPI composite hydrogels. All samples were stored at 4 °C overnight for further use.

### 3.3. The Properties of Sal–SPI Composite Hydrogel

#### 3.3.1. Apparent State

Briefly, different Sal–SPI composite hydrogel samples (SS0, SS4, SS5, SS6, SS7, SS8) were prepared and compared in appearance characteristics (formability, softness, surface smoothness, etc.).

#### 3.3.2. Test on Water Holding Capacity (WHC) and Water Content (WC)

WHC of Sal–SPI hydrogels was tested by centrifugal weighing method [43]. All samples (5.0 g) were centrifuged in an ultrafiltration centrifuge tube (5000 r/min, 40 min, 4 °C), then the isolated water was discarded. Subsequently, the samples were weighed and centrifuged again until the difference between the two weights was less than 0.05 g. The WHC was calculated by Equation (1).
(1)$$WHC\%=\frac{W_{1}-W_{0}}{W_{2}-W_{0}}*100\%$$
where W_0_ (g) was the weight of ultrafiltration centrifuge tube and W_1_ (g) was the total weight of ultrafiltration centrifuge tube and samples after centrifuging and W_2_ (g) was the total weight of ultrafiltration centrifuge tube and samples before centrifuging.

The WC was determined in a drying oven maintained at 80 °C (with air circulation). All hydrogel samples were cut into small pieces of 2 cm × 2 cm × 3 mm and weighed (m0), then they were dried in an oven at 80 °C to constant weight (m1). The WC of hydrogels was calculated by Equation (2).
(2)$$WC\%=\frac{m_{0}-m_{1}}{m_{0}}*100\%$$

#### 3.3.3. Investigation of Swelling Behavior

The ESD of the hydrogels was determined based on the method described by Richa et al. [44] with slight modifications. Freeze-dried hydrogel samples (2 cm × 2 cm × 3 mm) were weighed (W_0_) and dispersed in 0.9 wt% NaCl solution at 37 °C. The swollen hydrogels were taken out per hour and excessive water was wiped, then weighed and recorded until the weight of hydrogel samples (W_1_) no longer changed. The ESD of hydrogels was calculated by Equation (3).
(3)$$ESD\%=\frac{W_{1}-W_{0}}{W_{0}}*100\%$$

#### 3.3.4. Texture Properties Analysis

The texture properties of all samples were measured using a TMS-Pro texture analyzer (TMS-Pro, Virginia, VA, USA). The samples stored at 4 °C were taken out and made into a cylindrical shape with a height of 15 mm and a diameter of 20 mm. Then they were placed at room temperature for 1 h before testing. A P/20a cylindrical probe with a diameter of 20 mm was selected for testing. The test parameters are as follows: force sensing element range: 200 N, detection speed: 60 mm/min, deformation percentage: 30%, initial force: 0.4 N. The hardness, elasticity, and cohesion of each sample were obtained from the TPA curve.

#### 3.3.5. Scanning Electron Microscopy (SEM)

All samples were first broken by liquid nitrogen and then lyophilized by freeze vacuum dryer. The microstructures of the hydrogels were acquired using SEM (Nova NanoSEM 450, FEI, Hillsboro, OR, USA). The conductivity of the samples was increased by coating a thin layer of gold, then the brittle section of each sample was observed at 20 kV [45].

### 3.4. Rheological Measurements

The rheological measurements were carried out using rheometer (Haake Mars 3, Thermofisher, Hennigsdorf, Germany). All sample solutions were performed on strain scan to determine the strain within the linear viscoelastic range before performing the temperature-time scan. A parallel plate with a diameter of 35 mm, a gap of 1 mm, and a frequency of 1 Hz was applied for the measurements.

Heating and cooling procedures: The temperature control program was set to equilibrate at 25 °C for 300 s, then increased to 90 °C at a rate of 5 °C/min for 30 min, and then cooled down to 25 °C at a rate of −5 °C/min.

Heat preservation procedures: The temperature control program was set to equilibrate at 25 °C for 300 s, then increased to 50 °C at a rate of 5 °C/min for 60 min, and then cooled down to 25 °C at a rate of −5 °C/min.

The gelation process of Sal single hydrogel was performed with heating and cooling procedures and heat preservation procedures on the rheometer. Then TGase (20 U/g) was added to Sal single solution (2 mL) for heat preservation procedures and SPI single solution underwent heating and cooling procedures. The Sal–SPI composite solution (2 mL) underwent heating and cooling procedures firstly. Then the Sal–SPI composite solution was heated in a 90 °C water bath for 30 min and immediately cooled down to room temperature by running water, after which TGase was added into the composite solution for heat preservation procedures.

### 3.5. Spectral Analysis

#### 3.5.1. FTIR

An infrared spectrometer (TENSOR27, Karlsruhe, Germany) was applied for the infrared spectral analysis. Lyophilized different hydrogel samples were mixed with a certain amount KBr powders to be scanned. The spectra were recorded in the wavelengths ranging from 400 cm^−1^ to 4000 cm^−1^ at a resolution of 4 cm^−1^ [46].

#### 3.5.2. X-ray Diffraction (XRD)

X-ray diffractometer (XRD, Ultima IV, Tokyo, Japan) was used to study the crystal structure of Sal, SPI, and the prepared Sal–SPI composite hydrogel. The sample measurement conditions were as follows: 20 °C, 40 kV, 40 mA, diffraction width DS = SS = 1°, RS = 0.3 mm, scanning angle (2θ) range 5~85°, step length 0.02°, step speed 6°/min. The spectrum was analyzed and processed by MDI Jade software [40].

### 3.6. Biodegradability of Sal–SPI Composite Hydrogel

The biodegradability testing of the hydrogels was determined based on the method described by Das et al. [47] with slight modifications. Briefly, pre-weighed swollen samples were soaked in 30 mL of PBS buffer (0.1 M, pH 7.4) containing lysozyme (2 mg/mL) and incubated at 37 °C with continuous shaking. The hydrogel samples were carefully removed from the PBS and weighed every 12 h, then replaced with fresh PBS buffer. Calculate the remaining weight of the Sal–SPI composite gel using the following formula:(4)$$Remaining\;weight\left(\%\right)=\frac{W_{t}}{W_{0}}*100\%$$
where W_0_ is the initial weight of the swollen hydrogel, and Wt is the weight of the hydrogel at different time.

### 3.7. Statistical Analysis

SPSS Statistics 20.0 software was employed to analyze the data, and one-way analysis of variance (ANOVA) was used to determine the significant difference between each test (*p* < 0.05). All samples were performed in triplicate unless otherwise specified and all results were expressed as a mean ± standard deviation.

## 4. Conclusions

As the research has demonstrated, Sal–SPI composite hydrogels were successfully prepared by thermal treatment and TGase induction in this study. Furthermore, the apparent state, WHC, WC, swelling properties, texture characteristics, and microstructure of Sal–SPI composite hydrogels were found to be significantly influenced by Sal concentration. When Sal concentration varied from 0.4 to 0.6 wt%, the Sal–SPI composite hydrogel exhibited excellent moldability and the microscopic network structure of the hydrogel began to become homogeneous and dense. In addition, Sal and SPI can be cross-linked by hydrogen bonds to form a network structure, the thermal stability of the Sal–SPI composite hydrogel remarkably improved compared with the SPI single hydrogel. This research could provide an approach to developing Sal–SPI composite hydrogels and broaden the variety of applications for Sal and SPI.

## Data Availability

The data are available from the corresponding author.

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
