# Peer review of "Preparation and Properties of Salecan–Soy Protein Isolate Composite Hydrogel Induced by Thermal Treatment and Transglutaminase"

_ijms, 2022, doi:10.3390/ijms23169383_

Round 1

Reviewer 1 Report

Dear authors

This is an interesting research that performs a composite hydrogel based on a combination of Salecan and Soy Protein using thermal treatment and transglutaminase induction. This research only concentrates on the characterizations and properties of the hydrogel material. However, it should be significantly upgraded for publication.

1. Figure 5 are very poor. Please provide a clear figure

2. The introduction section did not present a significance and novelty of this research

3. Hydrogel is a well-known material and many reseaches have published  in this topic. Thus, a comparison of properties between this research and other published reseaches should be sumarrized in a table

4. Interacting mechanism and bonds between of Salecan and Soy Protein in the hydrogel formation must be expressed by a schemetic illustration

5. For evaluating applicatbility of the hydrogel, more characterizations should be conducted such as mechanical property tests and biocompatibility

Reviewer 2 Report

Please, see attached file.

Round 2

Reviewer 1 Report

The authors addressed all of my concerns. It can be accepted for publication